# Tomographic Auto-Encoder: Unsupervised Bayesian Recovery of Corrupted Data

**Francesco Tonolini**
School of Computing Science
University of Glasgow
Glasgow, UK
2402432t@student.gla.ac.uk

**Pablo Garcia Moreno**
Amazon
London, UK
morepabl@amazon.co.uk

**Andreas Damianou**
Amazon
London, UK
damianou@amazon.co.uk

**Roderick Murray-Smith**
School of Computing Science
University of Glasgow
Glasgow, UK
roderick.murray-smith@glasgow.ac.uk

## Abstract

We propose a new probabilistic method for unsupervised recovery of corrupted data. Given a large ensemble of degraded samples, our method recovers accurate posteriors of clean values, allowing the exploration of the manifold of possible reconstructed data and hence characterising the underlying uncertainty. In this setting, direct application of classical variational methods often gives rise to collapsed densities that do not adequately explore the solution space. Instead, we derive our novel *reduced entropy condition* approximate inference method that results in rich posteriors. We test our model in a data recovery task under the common setting of missing values and noise, demonstrating superior performance to existing variational methods for imputation and de-noising with different real data sets. We further show higher classification accuracy after imputation, proving the advantage of propagating uncertainty to downstream tasks with our model.

## 1 Introduction

Data sets are rarely clean and ready to use when first collected. More often than not, they need to undergo some form of pre-processing before analysis, involving expert human supervision and manual adjustments (Zhou et al., 2017; Chu et al., 2016). Filling missing entries, correcting noisy samples, filtering collection artefacts and other similar tasks are some of the most costly and time consuming stages in the data modeling process and pose an enormous obstacle to machine learning at scale (Munson, 2012). Traditional data cleaning methods rely on some degree of supervision in the form of a clean dataset or some knowledge collected from domain experts. However, the exponential increase of the data collection and storage rates in recent years, makes any supervised algorithm impractical in the context of modern applications that consume millions or billions of datapoints. In this paper, we introduce a novel variational framework to perform automated data cleaning and recovery without any example of clean data or prior signal assumptions.

The Tomographic auto-encoder (TAE), is named in analogy with standard tomography. Tomographic techniques for signal recovery aim at reconstructing a target signal, such as a 3D image, by algorithmically combining different incomplete measurements, such as 2D images from different view points, subsets of image pixels or other projections (Geyer et al., 2015). The TAE extends this concept to the reconstruction of data manifolds; our target signal is a clean data set, where corrupted data is interpreted as incomplete measurements. Our aim is to combine these to reconstruct the clean data.

More specifically, we are interested in performing Bayesian recovery, where we do not simply transform degraded samples into clean ones, but recover probabilistic functions, with which we can generate diverse clean signals and capture uncertainty. Uncertainty is considerably important when cleaning data. If we are over-confident about specific solutions, errors are easily ignored and passed

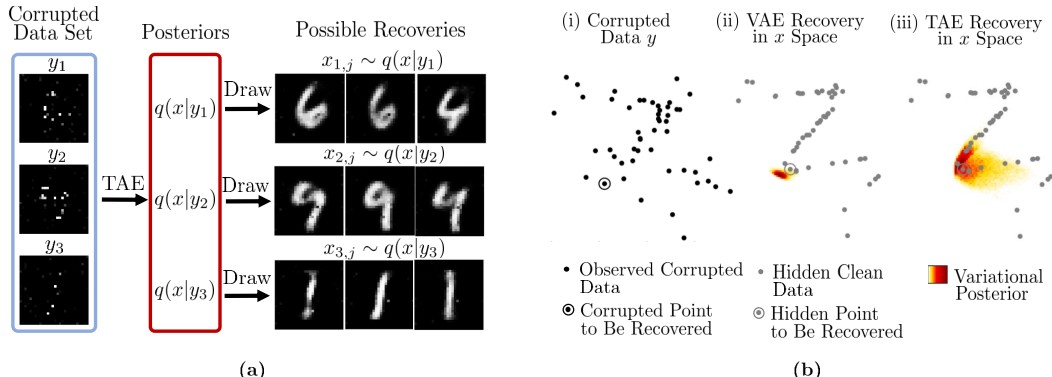

Figure 1: **(a)** Example of Bayesian recovery from corrupted data with a Tomographic Auto-Encoder (TAE) on corrupted MNIST. The TAE recovers posterior probability densities $q(x|y_i)$ for each corrupted sample $y_i$. We can draw from these to explore different possible clean solutions. **(b)** Two dimensional Bayesian recovery experiment. (i) Observed set of corrupted data $Y$, with the point we are inferring from $y_i$ highlighted. (ii) Ground truth hidden clean data with the target point $x_i$ highlighted, along with the posterior $q(x|y_i)$ reconstructed by a VAE. (iii) Posterior $q(x|y_i)$ recovered with our TAE. While the VAE posterior collapses to a single point, the TAE reconstructs a rich posterior that adjusts to the data manifold.

on to downstream tasks. For instance, in the example of figure 1(a), some corrupted observations are consistent with multiple digits. If we were to impute a single possibility for each sample, the true underlying solution may be ignored early on in the modeling pipeline and the digit will be consistently mis-classified. If we are instead able to recover accurate probability densities, we can remain adequately uncertain in any subsequent processing task.

Several variational auto-encoder (VAE) models have been proposed for applications that can be considered special cases of this problem (Im et al., 2017; Nazabal et al., 2018; Ainsworth et al., 2018) and, in principle, they are capable of performing Bayesian reconstruction. However, we show that surrogating variational inference (VI) in a latent space with VAEs results in collapsed distributions that do not explore the different possibilities of clean samples, but only return single estimates. The TAE performs approximate VI in the space of recovered data instead, through our *reduced entropy condition* method. The resulting posteriors adequately explore the manifold of possible clean samples for each corrupted observation and, therefore, adequately capture the uncertainty of the task.

In our experiments we focus on data recovery from noisy samples and missing entries. This is one of the most common data corruption settings being encountered in a wide range of domains with different types of data (White et al., 2011; Kwak & Kim, 2017). By testing our approach in this prevalent scenario, we can closely compare with recently proposed VAE approaches (Nazabal et al., 2018; Dalca et al., 2019; Mattei & Frellsen, 2019). We show how the existing VAE models exhibit the posterior collapse problem while the TAE produces rich posteriors that capture the underlying uncertainty. We further test TAEs on classification subsequent to imputation, demonstrating superior performance to existing methods in these downstream tasks. Finally, we use a TAE to perform automated missing values imputation on raw depth maps from the NYU rooms data set.

## 2 Method

In order to frame the problem and understand the issues with standard variational methods in this context, we view the task from a signal reconstruction prospective. The final scope of a Bayesian data recovery method is to build and train a parametric probability density function (PDF) $q(x|y)$, which takes as inputs corrupted samples $y$ and generates different possible corresponding clean data $x \sim q(x|y)$ through sampling. There are two aspects we need to design: i) the structure of this conditional PDF and ii) the way it will be trained to perform the recovery task.

Regarding the former, as natural data often lies on highly non-linear manifolds, we need the conditional PDF to capture complicated modalities, e.g. the distribution of plausible images consistent with one of the corrupted observations in figure 1(a). A suitable recovery PDF $q(x|y)$ needs to be able to capture such complexity. A natural choice to achieve high capacity and tractability is to

Figure 2: Training LVMs for data recovery. **(a)** Structure of the reconstruction LVM used to infer approximate posteriors $q(x|y)$ of clean data $x$ from corrupted observations $y$ as conditional inputs. **(b)** Training of $q(x|y)$ using a VAE. A prior in the latent space $z$ is introduced as a regulariser, however no explicit regularisation is imposed in $x$. **(c)** Training of $q(x|y)$ using our TAE model. An empirical prior $p(x) = \int p(z_p)p(x|z_p)dz_p$ is instead introduced in clean data space $x$.

construct $q(x|y)$ as a conditional latent variable model (LVM). Conditional LVM neural networks have achieved efficient and expressive variational inference in many recovery settings, capturing complex solution spaces in high dimensional problems, such as image reconstruction (Nguyen et al., 2017; Mirza & Osindero, 2014; Adler & Öktem, 2018). The conditional LVM consists of a first conditional distribution $q(z|y)$ mapping input corrupted data $y$ to latent variables $z$, and a second inference $q(x|z,y)$ mapping latent variables to output clean data $x$. The resulting PDF is $q(x|y) = \int q(z|y)q(x|z,y)dz$, where both $q(z|y)$ and $q(x|z,y)$ are simple distributions, such as isotropic Gaussians, whose moments are inferred by neural networks taking the respective conditional arguments as inputs. Figure 2(a) shows a graphical model for the conditional LVM.

While the choice of structure is fairly straightforward, the main difficulty lies in training the recovery LVM in the absence of clean ground truths $x$. In the supervised case, several established methods exist; the observed distributions of clean data $x$ conditioned on paired observations $y$ can be matched by parametric ones through a VAE or GAN training strategy (Sohn et al., 2015; Adler & Öktem, 2018; Tonolini et al., 2020). However, we are instead interested in the unsupervised situation, where we only have corrupted data $Y = \{y_{1:N}\}$ and a functional form for the corrupted data likelihood $p(y|x)$, e.g., missing values and additive noise. Training a conditional LVM to fit posteriors without any ground truth examples $x$ is rather challenging, as we do not have data to encode from, in the case of VAE architectures, or adversarially compare with, in the case of GAN models.

## 2.1 VAEs and the Posterior Collapse Problem

Variational auto-encoders (VAEs) have been proposed for several problems within this definition of unsupervised reconstruction (Dalca et al., 2019; Im et al., 2017; Ainsworth et al., 2018). These methods lead to good single estimates of the underlying targets. However, they easily over-fit their posteriors resulting in collapsed PDFs $q(x|y)$. Put differently, they are often unable to explore different possible solutions to the recovery problem and return single estimates instead. Figure 1(b-ii) shows this pathology in a two dimensional experiment.

The reason for this can be explained considering what the reconstruction LVM $q(x|y)$ is and how it is trained when directly employing a VAE in the unsupervised recovery scenario. The VAE encodes latent vectors $z$ from corrupted observations $y$ with an encoder $q(z|y)$ and reconstructs clean data $x$ with a decoder $p(x|z)$. These two functions constitute the reconstruction LVM $q(x|y) = \int q(z|y)p(x|z)dz$. As we do not have clean ground truths $x$, data likelihood is maximised by mapping reconstructed clean samples $x$ back to corrupted samples $y$ with a corruption process likelihood $p(y|x)$, e.g. zeroing out missing entries, to maximise reconstruction of the observations $y$. Concurrently, regularisation in the latent space is induced with a user defined prior $p(z)$ (e.g. a unit Gaussian). The resulting lower bound to be maximised during training can be expressed as follows:

$$\mathcal{L}_{VAE} = \mathbb{E}_{q(z|y)} \log p(y|z) - KL(q(z|y)||p(z)), \tag{1}$$

where the observations likelihood is $p(y|z) = \int p(x|z)p(y|x)dx$ and in some cases, such as for missing values and additive noise, it is analytical. A derivation is given in supplementary A.1.

Viewing the VAE training from a signal reconstruction prospective, where our reconstruction model is $q(x|y) = \int q(z|y)p(x|z)dz$, we can see that we are not introducing any prior directly on the hidden targets $x$, but only in the LVM latent space $z$. While regularising only in z may be computationally desirable, if the decoder $p(x|z)$ is of sufficient capacity, the model can learn to collapse regularised distributions in $z$ to single estimates in $x$, failing to capture uncertainty. In fact, this is induced by the objective function of equation 1; the model finds broad distributions in the latent space $q(z|y)$, which minimise the KL divergence with $p(z)$, but the generator $p(x|z)$ can learn to collapse them back to single maximum likelihood solutions in $x$, maximising $\mathbb{E}_{q(z|y)} \log \int p(x|z)p(y|x)dx$. This effect may be counteracted by reducing the capacity of $p(x|z)$ or the dimensionality of $z$, but doing so also reduces the capacity of the reconstruction model $q(x|y)$, resulting in an undesirable coupling between regularisation and posterior capacity.

## 2.2 SEPARATING POSTERIOR AND PRIOR: THE TOMOGRAPHIC AUTO-ENCODER

The premise of our model to address the aforementioned problem is simple: Introduce a prior $p(x)$ in the hidden clean signal space. In particular, we propose to use an empirical prior, having itself the form of an LVM $p(x) = \int p(z_p)p(x|z_p)dz_p$. In this way, we perform approximate variational inference in clean data space $x$, instead of surrogating it to the reconstruction function's latent space $z$. By doing so, we can control the capacity of the prior $p(x)$ to induce regularisation independently of the capacity of our reconstruction model $q(x|y) = \int q(z|y)q(x|z,y)dz$. For this framework, We can formulate the following ELBO:

$$\mathcal{L}_{TAE} = \mathbb{E}_{q(x|y)} \log p(y|x) + \mathbb{E}_{q(x|y)} \big[ \mathbb{E}_{q(z_p|x)} \log p(x|z_p) - KL(q(z_p|x)||p(z_p)) \big] + H(q(x|y)).$$

The above ELBO is derived in detail in supplementary section A.2. The main technical challenge and focus of this paper is how to compute and maximise the self entropy of the approximate posterior $H(q(x|y))$, as this conditional distribution is an LVM of the form $q(x|y) = \int q(z|y)q(x|z,y)dz$.

**Reduced Entropy Condition:** Direct computation of the entropy of an LVM model $q(x|y) = \int_z q(z|y)q(x|z,y)dz$ is intractable in the general case. Titsias & Ruiz (2019) proposed an approximate inference method to compute the gradient of the LVM's entropy for variational inference in latent spaces. However, this involves multiple samples to be drawn and evaluated with the LVM, which is expected to scale in complexity as the dimensionality and capacity of the target distribution increase.

In our case, we aim to approximately compute and optimise the entropy $H(q(x|y))$ for a distribution capturing natural data, which can be high-dimensional and lie on complicated manifolds. In order to maintain efficiency in the entropy estimation, we propose a new strategy; we identify a class of LVM posteriors for which the entropy reduces to a tractable form and then approximately constrain the posterior to such a class in our optimisation. Our main result is summarized in the following theorem:

**Theorem 1** *If $\frac{q(z|x,y)}{q(z|y)} = B\delta(z - g(x,y))$, where $\delta(\cdot)$ is the Dirac Delta function, $B$ is a real positive parameter and $g(x,y)$ is a deterministic function, then $H(q(x|y)) = H(q(z|y)) + \mathbb{E}_{q(z|y)}H(q(x|z,y))$.*

We detail the proof in supplementary A.3. Theorem 1 states that if the posterior over latent variables $q(z|x,y)$ is infinitely more localised than the latent conditional $q(z|y)$, then the LVM entropy $H(q(x|y))$ has the tractable form given above. This condition imposes the LVM posterior to present non-overlapping conditionals $q(x|z,y)$ for different latent variables $z$, but does not impose any explicit restriction to the capacity of the model. We can also formulate the condition as follows:

$$\mathbb{E}_{q(x,z|y)} \log \frac{q(z|x,y)}{q(z|y)} = C, \quad C \to \infty. \tag{2}$$

The proof is provided in supplementary section A.4. To train our posterior $q(x|y)$, we aim to maximise the ELBO $\mathcal{L}_{TAE}$ with the reduced entropy, while enforcing the condition of equation 2:

$$\arg\max \quad \mathbb{E}_{q(x|y)} \log p(y|x) + \mathbb{E}_{q(x|y)} \big[ \mathbb{E}_{q(z_p|x)} \log p(x|z_p) - KL(q(z_p|x)||p(z_p)) \big]$$

$$+ H(q(z|y)) + \mathbb{E}_{q(z|y)}H(q(x|z,y)), \quad s.t. \quad \mathbb{E}_{q(x,z|y)} \log \frac{q(z|x,y)}{q(z|y)} = C, \quad C \to \infty. \tag{3}$$

While the ELBO is now amenable to stochastic optimization, the constraint is intractable since $C \to \infty$ and the posterior $q(z|x, y)$ is intractable.

**Relaxed Constraint:** To render the constraint tractable, we firstly relax $C$ to be a positive hyper-parameter. The higher the value of $C$, the more localised $q(z|x, y)$ is imposed to be compared to $q(z|y)$ and the closest the reduced entropy is to the true one.

To address the intractability of the posterior $q(z|x, y)$, we employ a variational approximation with a parametric function $r(z|x, y)$. In fact, for any valid probability density $r(z|x, y)$, we can prove that

$$\mathbb{E}_{q(x,z|y)} \log \frac{q(z|x, y)}{q(z|y)} \geq \mathbb{E}_{q(x,z|y)} \log \frac{r(z|x, y)}{q(z|y)}. \tag{4}$$

The proof is given in supplementary section A.5. The above bound implicates the following:

$$\mathbb{E}_{q(x,z|y)} \log \frac{r(z|x, y)}{q(z|y)} = C \Rightarrow \mathbb{E}_{q(x,z|y)} \log \frac{q(z|x, y)}{q(z|y)} \geq C.$$

This means that imposing the condition with a parametric distribution $r(z|x, y)$, which is trained along with the rest of the model, ensures deviation from the set condition only by excess. As the exact condition is met only at $\mathbb{E}_{q(x,z|y)} \log \frac{q(z|x)}{q(z|y)} \to \infty$, we can never relax the constraint more than already set by the finite value of $C$.

**The TAE Objective Function:** Having defined a tractable ELBO and a tractable condition, we need to perform the constrained optimisation

$$\arg \max \quad \mathbb{E}_{q(x|y)} \log p(y|x) + \mathbb{E}_{q(x|y)} \big[ \mathbb{E}_{q(z_p|x)} \log p(x|z_p) - KL(q(z_p|x)||p(z_p)) \big]$$
$$+ H(q(z|y)) + \mathbb{E}_{q(z|y)} H(q(x|z, y)), \quad s.t. \quad \mathbb{E}_{q(x,z|y)} \log \frac{r(z|x, y)}{q(z|y)} = C. \tag{5}$$

We use the commonly adopted penalty function method (Zangwill, 1967; Phuong et al., 2018) and relax equation 5 to an unconstrained optimisation with the use of a positive hyper-parameter $\lambda$:

$$\arg \max \quad \mathbb{E}_{q(x|y)} \log p(y|x) + \mathbb{E}_{q(x|y)} \big[ \mathbb{E}_{q(z_p|x)} \log p(x|z_p) - KL(q(z_p|x)||p(z_p)) \big]$$
$$+ H(q(z|y)) + \mathbb{E}_{q(z|y)} H(q(x|z, y)) - \lambda \left| \mathbb{E}_{q(z,x|y)} \log \frac{r(z|x, y)}{q(z|y)} - C \right|. \tag{6}$$

To train the model, we perform the maximisation of equation 6 using the ADAM optimiser. Once the model is trained, we can generate diverse reconstructions from a corrupt observation $y_i$ by sampling from the posterior $q(x|y_i)$. Details of our optimisation are reported in supplementary B.1. We describe how we handle parameters of the corruption process $p(y|x)$ in supplementary B.2.

## 3 RELATED WORK

### 3.1 SUPERVISED BAYESIAN RECONSTRUCTION

The reconstruction of posterior densities from incomplete measurements has been recently investigated in supervised situations, where examples of clean data are available. In particular, conditional generative models were demonstrated with high dimensional data (Parmar et al., 2018). These methods work by exploiting an LVM to generate diverse realisations of targets conditioned on associated observations (Isola et al., 2017; Nguyen et al., 2017). Both conditional generative adversarial networks (CGANs) (Mirza & Osindero, 2014; Isola et al., 2017) and conditional VAEs (CVAEs) (Sohn et al., 2015; Nguyen et al., 2017) have been studied in this context. In both cases, the samples generated by conditioning on an observation can be interpreted as samples from the corresponding conditional posterior densities.

These approaches proved successful in a range of recovery tasks: reconstruction of images with missing groups of pixels (Nguyen et al., 2017), super-resolution (Parmar et al., 2018), medical computed tomography reconstructions (Adler & Öktem, 2018) and semi supervised situations, where examples of clean data and conditions are available in different amounts (Kingma et al., 2014; Denton et al., 2016; Tonolini et al., 2020). Other works reconstruct manifolds of solutions from observations only, but can be considered supervised, as they exploit pre-trained generators (Anirudh et al., 2018).

These works make the important observation that when learning to recover data from corrupted or partial observations, there is not a single right solution, but many differently likely ones. We aim to extend this ability to completely unsupervised scenarios, where no clean data examples are available.

### 3.2 Unsupervised Bayesian Reconstruction

Reconstructing posteriors in the unsupervised case is largely still an open problem. However, several tasks that fall within this definition have been recently approached with Bayesian machine learning methods. Arguably the most investigated is de-noising. Several works solve this problem by exploiting the natural tendency of neural networks to regularise outputs (Lehtinen et al., 2018; Krull et al., 2019a;b). Other methods build LVMs that explicitly model the noise process in their decoder, retrieving clean samples upon encoding and generation (Im et al., 2017; Creswell & Bharath, 2018).

A second notable example is that of missing value imputation. Corrupted data corresponds to samples with missing entries. Recent works have explored the use of LVMs to perform imputation, both with GANs (Li et al., 2019; Yoon et al., 2018; Luo et al., 2018) and VAEs (Nazabal et al., 2018; Mattei & Frellsen, 2019; Ma et al., 2018). In the former, the discriminator of the GAN is trained to distinguish real values from imputed ones, such that the generator is induced to synthesise realistic imputations. In the latter, the encoder of a VAE maps incomplete samples to a latent space, to then generate complete samples. Successful unsupervised Bayesian missing value imputation has also been demonstrated with neural processes, where a global latent representation is learned to generate input-output models used to impute in each example (Garnelo et al., 2018).

Finally, Bayesian LVM methods have been used on other unsupervised tasks that can be cast as special cases of data recovery problems. Amongst these, we find Multi-view generation (Shang et al., 2017; Ainsworth et al., 2018), where the target clean data includes all views for each samples, but the observed data only presents subsets. Blind source separation can also be cast as a recovery problem and has been approached with GANs and VAEs (Kameoka et al., 2018; Hoshen, 2019).

These models proved to be successful at reconstructing data in their specific domain. However, in our work, we show how exploiting a standard VAE inference structure, similarly to several of the aforementioned methods, often leads to posteriors of clean data that collapse on single estimates, sacrificing the probabilistic capability of LVMs.

### 3.3 Posterior Collapse in Variational Inference

The posterior collapse problem we approach with the reduced entropy condition method presented in this paper has some analogy with the latent posterior collapse encountered when using implicit distributions in variational inference to obtain flexible recognition models. The main issue in training these models successfully without collapse is the computation of density rations between the latent prior and the implicit variational posterior. This problem is analogous to the difficulty in estimating the LVM entropy in our method. Yin & Zhou (2018) proposed to use a further lower bound on the ELBO and add a term encouraging diversity to avoid collapse. This term is obtained by drawing $K$ Gaussian components from the LVM posterior and computing the KL divergence of an individual component with the mixture distribution, i.e. the sum of the drawn Gaussians. Titsias & Ruiz (2019) build on this work by deriving an unbiased estimator for the ELBO gradient, instead of using a surrogate lower bound.

These methods rely on estimating the LVM posterior through sampling and aggregating $K$ explicit distribution components. This was demonstrated to work well for posteriors in artificial latent space by drawing only a few components. However, in our data recovery setting, we need to capture the posteriors in clean data space, rather than a latent space. Posteriors capturing the uncertainty in natural data are expected to be much more complex and higher dimensional, leading to the number $K$ of drawn Gaussians needed to approximate the true LVM with these methods to become rather large, making optimisation inefficient or even intractable in extreme cases. The reduced entropy condition method we derive in this paper avoids the posterior collapse without having to estimate the LVM posterior through sampling and is therefore specially suited for the data recovery setting, where we are required to capture posteriors in clean data space.

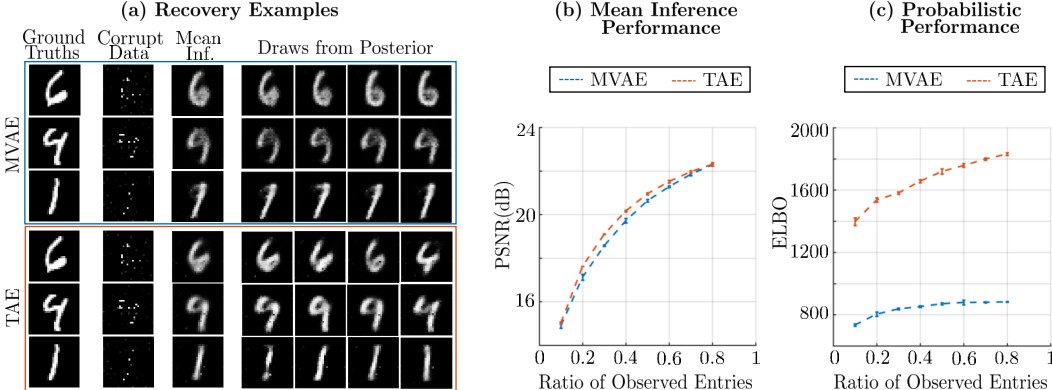

Figure 3: MNIST data recovery from missing entries and noise. **(a)** Recoveries using an MVAE and our TAE, showing average reconstruction and samples from the trained posteriors. **(b)** PSNR between ground truths and mean reconstruction. **(c)** ELBO assigned by the recovered posteriors to the ground truth data. The mean inference performance is very similar for the two models (PSNR values), while the probabilistic performance (ELBO values) is significantly higher for our TAE model. We can see evidence of this difference in the reconstruction examples. The MVAE and TAE return similarly adequate mean solutions, but the MVAE posterior's draws are all very similar, suggesting that the posterior has collapsed on a particular reconstruction. Contrarily, the posteriors returned by the TAE explore different possible solutions that are consistent with the associated corrupted observation.

Table 1: Bayesian recovery from noisy data with different percentages of missing entries. Table shows the ELBO assigned by the retrieved posteriors to the ground truth clean data. Our TAE model consistently returns higher ELBO values compared to the competing variational methods, as it is able to retrieve rich posteriors that adequately sample the solution space. More values in supp. D.3.

|  | MNIST | | Fashion-MNIST | | UCI HAR | |
|---|---|---|---|---|---|---|
|  | 50% | 80% | 50% | 80% | 50% | 80% |
| MVAE | $870 \pm 6$ | $803 \pm 15$ | $757 \pm 1$ | $723 \pm 7$ | $585 \pm 4$ | $471 \pm 10$ |
| MIWAE | $917 \pm 4$ | $780 \pm 6$ | $800 \pm 7$ | $766 \pm 8$ | $613 \pm 6$ | $584 \pm 8$ |
| TAE | $\mathbf{1719 \pm 7}$ | $\mathbf{1536 \pm 14}$ | $\mathbf{1326 \pm 7}$ | $\mathbf{1094 \pm 13}$ | $\mathbf{1014 \pm 6}$ | $\mathbf{854 \pm 52}$ |

## 4 EXPERIMENTS

### 4.1 POSTERIOR RECOVERY

We corrupt the MNIST dataset (Deng, 2012) by introducing missing values and additive Gaussian noise on the observed entry. We then train both a missing value imputation VAE (MVAE), analogous to those presented in (Nazabal et al., 2018) and (Dalca et al., 2019), and our TAE model with the corrupted data sets. The VAE and TAE are constructed such that the structure of their posteriors $q(x|y)$, i.e. the functions mapping corrupted data to distributions of clean data at test time, are exactly the same. In this way, we can ensure that differences in performance are due to the variational inference method employed and not the choice of posterior model. The resulting variational posteriors are used to perform data recovery from the corrupted samples. Fig. 3(a) shows examples of mean reconstruction and posterior draws. See analogous experiments for grouped missings in suppl. D.2.

We evaluate the accuracy of mean reconstruction at different ratios of observed entries by measuring the peak signal to noise ratio (PSNR) between the ground truth data and mean recoveries (Figure 3(b)). To evaluate probabilistic performance we approximately measure the likelihood assigned by the recovered posteriors to the ground truth data through a reconstruction ELBO, by training a new inference function with the clean ground truths, but leaving the posterior fixed, as is common for evaluating ELBOs in unsupervised settings (Cremer et al., 2018; Mattei & Frellsen, 2018; 2019). A detailed description of this approach is given in supplementary C.3. We also carry out analogous experiments testing de-noising and missing value imputation separately. THese results are reported in supplementary D.4 and D.5. Results are shown in figure 3(c). We further evaluate our TAE with Fashion-MNIST – $28 \times 28$ grey-scale images of clothing (Xiao et al., 2017), and the UCI HAR dataset, which consists of filtered accelerometer signals from mobile phones worn by different people

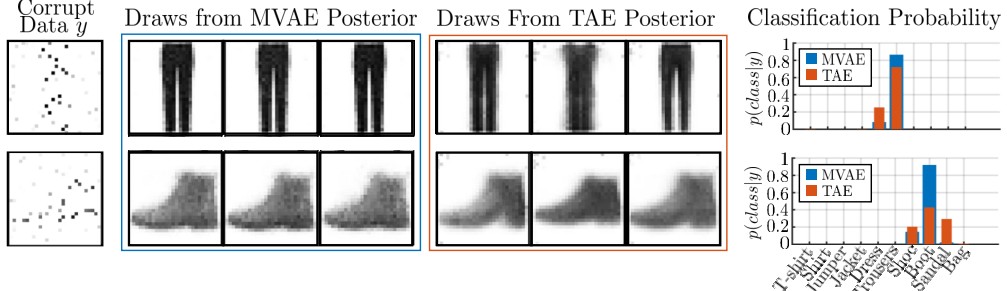

Figure 4: Propagating uncertainty to a classification task. Draws from the MVAE posterior are all very similar to each other. As a result, the imputed images are almost always classified in the same way and the uncertainty of the task is underestimated. The TAE posterior explores varied possible solutions to the recovery task. These can be recognised as different classes, resulting in less concentrated distributed probabilities that better reflect the associated uncertainty.

during common activities (Anguita et al., 2012). As before, we test the recovery of these data sets from a version affected by missing values and additive noise. In addition to the MVAE baseline, we compared against the recently proposed missing values importance weighted auto encoder (MIWAE) (Mattei & Frellsen, 2019), which optimises an importance weighted ELBO in place of the standard one. For each model and settings we compute the ELBO assigned to the ground truth data. Results are shown in Table 2. Experimental details in Sec. C of suppl. mat.

## 4.2 DOWNSTREAM TASKS

To investigate the advantage of capturing complex uncertainties with our TAE model, we are interested in testing performance in downstream tasks. We test classification performance on subsets of the MNIST and Fashion-MNIST data sets, after recovery with our TAE. With both sets, we consider situations in which 10.000 examples are available, but corrupted with missing entries and noise. 1,000 of these are labelled with one of 10 possible classes and we wish to classify the remaining 9,000. To do so, we first train the TAE model on the full set, then use the recovered posteriors to generate multiple possible cleaned data for the labelled sub-set and use them to train a classifier.

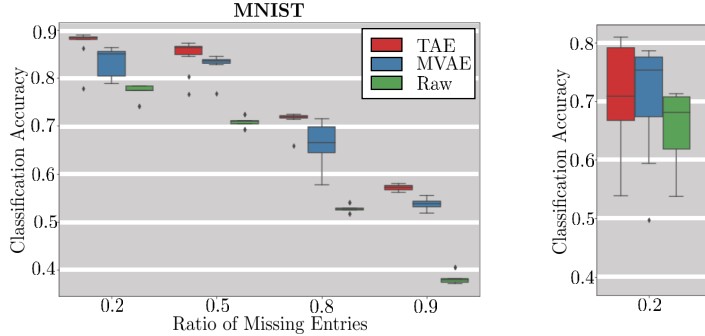
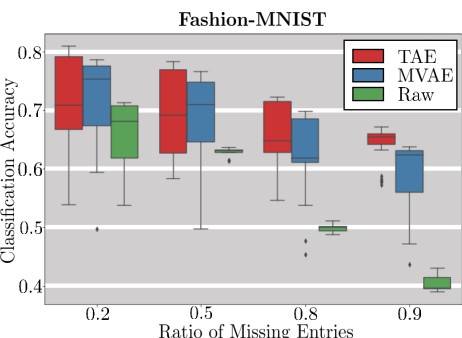

Figure 5: Classification accuracy after imputation. Classifying using TAE imputations gives an advantage in this downstream task over using raw corrupted data and MVAE imputations, especially when the number of missing entries is high. This is because the MVAE collapses on single imputations, while the TAE generates diverse samples for each corrupted observation. The TAE classifier trains with data augmentations consistent with observed corrupted images, instead of single estimates.

To perform classification on the 9,000 remaining examples, we generate multiple possible cleaned data with the variational posteriors. Then, for each posterior sample, we perform classification and histogram the results. Examples are shown in figure 4. To evaluate the performance, we take the class with the largest histogram as the inferred one. We repeat this experiment for different ratios of

missing values and several repetitions, varying the subsets of labelled and unlabelled data to be used. Classification accuracy results are shown in figure 5.

### 4.3 MISSING VALUES IN THE NYU DEPTH MAPS

As a final practical application, we use a convolutional version of our TAE to perform structured missing value imputation on depth maps of indoors rooms collected with a Kinect depth sensor. Missing entries are very common in depth maps recorded with such structured light sensors (Scharstein & Szeliski, 2003). We use raw depth data from the NYU rooms dataset, commonly used to test various computer vision systems (Silberman & Fergus, 2011; Silberman et al., 2012; Dollár & Zitnick, 2013; Chang et al., 2018). A large portion of the set is available only as raw data, which presents missing entries. These are especially concentrated around objects' edges and reflecting surfaces, breaking the common assumption of missing at random, making this task particularly challenging. We train our TAE with a subset of this raw data set to perform imputation. Examples of results are shown in figure 11. Additional examples are shown in supplementary section D.6.

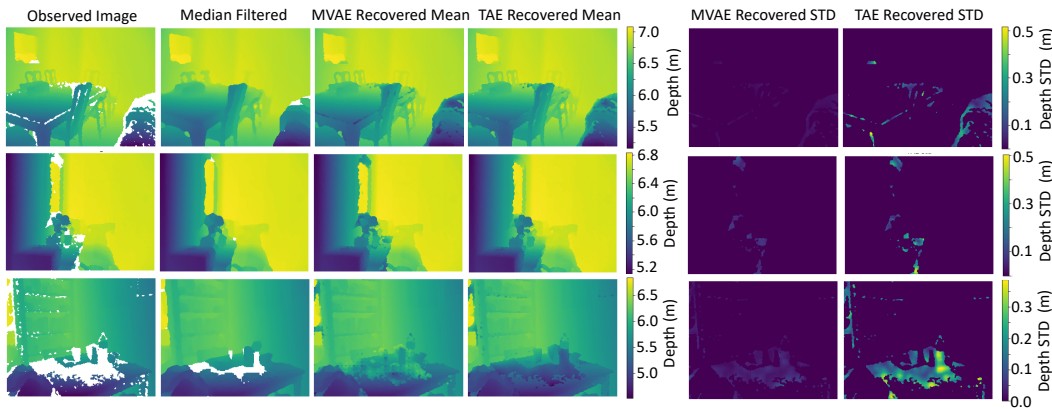

Figure 6: Unsupervised missing value imputation with our TAE on raw depth maps from the NYU rooms data set, compared with a median filter approach and the standard MVAE. Missing pixels in the observed images are in white. The median filter results in overly smoothed images and is unable to fill pixels that are surrounded by large missing areas. The MVAE returns adequate reconstructions, however, it over-fits to inaccurate solutions in certain locations, returning low uncertainty. The TAE returns good reconstructions and assigns high uncertainty to locations where reconstructions are most inaccurate, as shown by the marginal standard deviations.

## 5 CONCLUSION

We presented tomographic auto-encoders; a variational inference method for recovering posterior distributions of clean data from a corrupted data set alone. We derive the *reduced entropy condition* method; a novel inference strategy that results in rich distributions of clean data given corrupted observations, thereby capturing the uncertainty of the task, while standard variational methods often collapse on single answers. In our experiments, we demonstrate this capability and show the advantage of capturing uncertainty with the TAE in downstream tasks, outperforming the state-of-the-art VAE based recovery methods.

### ACKNOWLEDGEMENTS

F.T. and R.M-S. acknowledge funding support from Amazon and EPSRC grants EP/M01326X/1, EP/T00097X/1 *(QuantIC – UK Quantum Technology Hub in Quantum Enhanced Imaging)*. R.M-S. acknowledges funding support from EP/R018634/1 *(Closed-Loop Data Science for Complex, Computationally- and Data-Intensive Analytics)*.

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

# Tomographic Auto-Encoder - Supplementary material

## A PROOFS AND DERIVATIONS

### A.1 DERIVATION OF VAE ELBO FOR DATA RECOVERY

We aim to maximise the log likelihood of the observed corrupted data $y$

$$\log p(y) = \log \int_x \underbrace{\int_z p(z)p(x|z)dz}_{p(x)} \, p(y|x)dx. \tag{7}$$

We can introduce a variational distribution in both clean data space and latent space $q(x, z|y)$ and define a lower bound as

$$
\begin{aligned}
\log p(y) \geq & \int_x \int_z q(x, z|y) \log \frac{p(z)p(x|z)dz}{q(x, z|y)} dzdx \\
& + \int_x \int_z q(x, z|y) \log p(y|x)dzdx.
\end{aligned}
\tag{8}
$$

To obtain the VAE ELBO used in data recovery settings, the choice of the variational posterior is $q(x, z|y) = q(z|y)p(x|z)$. The ELBO can then be simplified to give

$$
\begin{aligned}
\log p(y) \geq & \int_x \int_z q(z|y)p(x|z) \log \frac{p(z)p(x|z)dz}{q(z|y)p(x|z)} dzdx \\
& + \int_x \int_z q(z|y)p(x|z) \log p(y|x)dzdx \\
= & \underbrace{\int_x p(x|z)dx}_{=1} \int_z q(z|y) \log \frac{p(z)dz}{q(z|y)} dz \\
& + \int_x \int_z q(z|y)p(x|z)dz \log p(y|x)dx
\end{aligned}
\tag{9}
$$

For situations in which the observations' likelihood $\int_x p(x|z)p(y|x)dx$ has a closed form, such as additive noise and missing entries, we can define a tighter bound to the likelihood by moving the integral in $x$ in the second term inside the logarithm

$$
\begin{aligned}
\log p(y) \geq & \int_z q(z|y) \log \frac{p(z)dz}{q(z|y)} dz \\
& + \int_z q(z|y) \log \left[ \int_x p(y|x)p(x|z)dx \right] dz \\
= & - KL(q(z|y)||p(z)) + \mathbb{E}_{q(z|y)} \log p(y|z)dx.
\end{aligned}
\tag{10}
$$

Because $p(x|z)$ simplifies in the KL term, this ELBO avoids variational inference in the space of clean data $x$.

### A.2 DERIVATION OF TAE ELBO

In our TAE model we defined separate LVMs for prior and posterior. To distinguish between the posterior latent variable and the prior latent variable, we name the former $z$ and the latter $z_p$. The likelihood we aim to maximise is

$$\log p(y) = \log \int_x \underbrace{\int_{z_p} p(z_p)p(x|z_p)dz_p}_{p(x)} \, p(y|x)dx. \tag{11}$$

Similarly to the VAE ELBO case, we define a variational posterior $q(x, z_p|y)$ to find a lower bound

$$
\begin{aligned}
\log p(y) \geq & \int_x \int_{z_p} q(x, z_p|y) \log \frac{p(z_p)p(x|z_p)dz_p}{q(x, z_p|y)} dz_p dx \\
& + \int_x \int_{z_p} q(x, z_p|y) \log p(y|x) dz_p dx.
\end{aligned}
\tag{12}
$$

However, in our model we do not make the assumption that the variational posterior has the special form described in section A.1 and instead set it to have the form $q(x, z_p|y) = q(x|y)q(z_p|x)$, separating posterior inference from observations $y$ to clean data $x$ and inference of prior latent variables $z_p$. The resulting lower bound is

$$
\begin{aligned}
\log p(y) \geq & \int_x \int_{z_p} q(x|y)q(z_p|x) \log \frac{p(z_p)p(x|z_p)}{q(x|y)q(z_p|x)} dz_p dx + \int_x \int_{z_p} q(x|y)q(z_p|x) \log p(y|x) dz_p dx \\
= & \int_x q(x|y) \underbrace{\int_{z_p} q(z_p|x) \log \frac{p(z_p)p(x|z_p)}{q(z_p|x)} dz_p}_{\geq \log p(x)} dx + \int_x \underbrace{\int_{z_p} q(z_p|x) \, dz_p}_{=1} q(x|y) \log p(y|x) dx \\
& - \int_x \underbrace{\int_{z_p} q(z_p|x) dz_p}_{=1} q(x|y) \log q(x|y) dx \\
= & \mathbb{E}_{q(x|y)} \big[ \mathbb{E}_{q(z_p|x)} \log p(x|z_p) - KL(q(z_p|x)||p(z_p)) \big] + \mathbb{E}_{q(x|y)} \log p(y|x) + H(q(x|y)).
\end{aligned}
\tag{13}
$$

### A.3 PROOF OF THEOREM 1

$$
\begin{aligned}
& \frac{q(z|x, y)}{q(z|y)} = B\delta(z - g(x, y)) \implies \frac{q(z|x, y)}{q(z|y)} \frac{q(z'|x, y)}{q(z'|y)} = 0, \quad \forall x, z \neq z' \\
\implies & \frac{q(x|z, y)}{q(x|y)} \frac{q(x|z', y)}{q(x|y)} = 0, \quad \forall x, z \neq z' \\
\implies & q(x|z, y)q(x|z', y) = 0, \quad \forall x, z \neq z' \\
\implies & q(x|z', y) = 0, \quad \forall x \sim q(x|z, y), z \neq z'
\end{aligned}
\tag{14}
$$

Using the result of equation 14, we can derive the form of the entropy $H(q(x|y))$ for this special case as the following:

$$
\begin{aligned}
H(q(x|y)) = & - \int_x \left[ \int_z q(z|y)q(x|z, y)dz \right] \cdot \log \left[ \int_{z'} q(z'|y)q(x|z', y)dz' \right] dx \\
= & - \int_x \int_z q(z|y)q(x|z, y) \cdot \log \Bigg[ \int_{z'=z} q(z'|y)q(x|z', y)dz' \\
& + \underbrace{\int_{z' \neq z} q(z'|y)q(x|z', y)dz'}_{eq.14 \implies = 0} \Bigg] dz dx \\
= & - \int_z \int_x q(z|y)q(x|z, y) \log [q(z|y)q(x|z, y)] \, dx dz \\
= & - \int_z \int_x q(z|y)q(x|z, y) \log q(z|y) dx dz - \int_z \int_x q(z|y)q(x|z, y) \log q(x|z, y) dx dz \\
= & - \int_z q(z|y) \log q(z|y) dz - \int_z q(z|y) \int_x q(x|z, y) \log q(x|z, y) dx dz \\
= & H(q(z|y)) + \mathbb{E}_{q(z|y)} H(q(x|z, y)).
\end{aligned}
\tag{15}
$$

## A.4    PROOF OF THE EQUIVALENCE BETWEEN CONDITIONS

**proof of necessary condition:**

$$
\begin{aligned}
\mathbb{E}_{q(x,z|y)} \log \frac{q(z|x,y)}{q(z|y)} &= \int_z q(z|y) \int_x q(x|z,y) \log \frac{q(z|x,y)}{q(z|y)} dxdz \\
&= \int_x q(x|y) \int_z q(z|x,y) \log \frac{q(z|x,y)}{q(z|y)} dzdx \\
&= \int_x q(x|y) \int_z q(z|x,y) \log q(z|x,y) dzdx \\
&\quad - \int_x q(x|y) \int_z q(z|x,y) \log q(z|y) dzdx \\
&= \int_x q(x|y) \int_z q(z|x,y) \log q(z|x,y) dzdx \\
&\quad - \underbrace{\int_x q(x|z,y) dx}_{=1} \int_z q(z|y) \log q(z|y) dz \\
&= \mathbb{E}_{q(x|y)} \underbrace{\int_z q(z|x,y) \log q(z|x,y) dz}_{-H(q(z|x,y))} \\
&\quad - \underbrace{\int_z q(z|y) \log q(z|y) dz}_{-H(q(z|y))} .
\end{aligned}
\tag{16}
$$

If the above expression tends to infinity, either $H(q(z|x,y)) \to -\infty$ or $H(q(z|y)) \to \infty$, meaning that either $q(z|x,y) \to$ a Delta function, or $q(z|y) \to$ uniform. Either condition implies $\frac{q(z|x,y)}{q(z|y)} = B\delta(z - g(x,y))$.

**proof of sufficient condition:**

$$
\begin{aligned}
\mathbb{E}_{q(x,z|y)} \log \frac{q(z|x,y)}{q(z|y)} &= \int_x q(x|y) \int_z q(z|x,y) \log \frac{q(z|x,y)}{q(z|y)} dzdx \\
&= \int_x q(x|y) \int_z q(z|y) \frac{q(z|x,y)}{q(z|y)} \log \frac{q(z|x,y)}{q(z|y)} dzdx.
\end{aligned}
\tag{17}
$$

Now we set $\frac{q(z|x,y)}{q(z|y)} = B\delta(z - g(x,y))$:

$$
\begin{aligned}
&\int_x q(x|y) \int_z q(z|y) B\delta(z - g(x,y)) \log B\delta(z - g(x,y)) dzdx \\
&= \int_x q(x|y) q(g(x,y)|y) \log B \underbrace{\delta(g(x,y) - g(x,y))}_{\to \infty, \forall x} dx.
\end{aligned}
\tag{18}
$$

Therefore, $\frac{q(z|x,y)}{q(z|y)} = B\delta(z - g(x,y))$ is a sufficient condition for $\mathbb{E}_{q(x,z|y)} \log \frac{q(z|x,y)}{q(z|y)} \to \infty$.

## A.5 PROOF OF EQUATION 6

$$
\begin{aligned}
\mathbb{E}_{q(x,z|y)} \log \frac{q(z|x,y)}{q(z|y)} &= \int_z \int_x q(x,z|y) \log q(z|x,y) dz dx \\
&\quad - \int_z \int_x q(x,z|y) \log q(z|y) dz dx \\
&= \int_x q(x|y) \int_z q(z|x,y) \log q(z|x,y) dz dx \\
&\quad - \int_z \int_x q(x,z|y) \log q(z|y) dz dx \\
&\geq \int_x q(x|y) \int_z q(z|x,y) \log r(z|x,y) dz dx \\
&\quad - \int_z \int_x q(x,z|y) \log q(z|y) dz dx \\
&= \mathbb{E}_{q(x,z|y)} \log \frac{r(z|x,y)}{q(z|y)},
\end{aligned}
\tag{19}
$$

Where the inequality derives from the positivity of the KL divergence $KL(q(z|x,y)||r(z|x,y))$.

## B ALGORITHM

### B.1 DETAILS OF TRAINING

As detailed in section 3.2, to train our variational posterior $q(x|y)$, we maximise through gradient ascent the TAE ELBO with the reduced entropy penalty function

$$
\begin{aligned}
\arg\max \quad & \mathbb{E}_{q(x|y)} \log p(y|x) + \mathbb{E}_{q(x|y)} \big[ \underbrace{\mathbb{E}_{q(z_p|x)} \log p(x|z_p) - KL(q(z_p|x)||p(z_p))}_{Prior \quad ELBO, \quad \geq p(x)} \big] \\
& + H(q(z|y)) + \mathbb{E}_{q(z|y)} H(q(x|z,y)) - \lambda \left| \mathbb{E}_{q(z,x|y)} \log \frac{r(z|x,y)}{q(z|y)} - C \right|.
\end{aligned}
\tag{20}
$$

All expectations in the above expression are computed and optimised by sampling the corresponding conditional distributions using the re-parametrisation trick characteristic of VAEs.

Because the prior LVM $p(x) = \int p(z_p) p(x|z_p) dz_p$ is training entirely with samples from the posterior LVM, which is also training, the model can easily obtain high values for the prior ELBO by generating collapsed samples $x$ with the posterior and get stuck in an unfavourable local minimum. TO avoid this, we employ a warm up strategy. We define a positive parameter $\gamma$ that multiplies the expectation of the prior ELBO and the entropy $H(x|z,y)$:

$$
\begin{aligned}
\arg\max \quad & \mathbb{E}_{q(x|y)} \log p(y|x) + \gamma \mathbb{E}_{q(x|y)} \big[ \underbrace{\mathbb{E}_{q(z_p|x)} \log p(x|z_p) - KL(q(z_p|x)||p(z_p))}_{Prior \quad ELBO, \quad \geq p(x)} \big] \\
& + H(q(z|y)) + \gamma \mathbb{E}_{q(z|y)} H(q(x|z,y)) - \lambda \left| \mathbb{E}_{q(z,x|y)} \log \frac{r(z|x,y)}{q(z|y)} - C \right|.
\end{aligned}
\tag{21}
$$

The value of $\gamma$ is initially set to zero. After a set number of iterations it is linearly increased to reach one and kept constant for the remaining training iterations.

### B.2 COMPLETE OBJECTIVE FUNCTION

**Observation Parameters:** In the general case, the corruption process $p(y|x)$, mapping clean data $x$ to degraded samples $y$, is controlled by parameters that differ from sample to sample. We can distinguish these into observed parameters $\alpha$ and unobserved parameters $\beta$. For example, in the case of missing values and noise, the indexes of missing entries in each sample are often observed parameters, while the noise level is an unobserved parameter. The complete form of the corruption likelihood for a clean sample $x_i$ is then $p(y|x_i, \alpha_i, \beta_i)$.

**Objective Function:** With the parameters conditionals described in subsection 3.2.4 and explicitly showing the parameters to be optimised, the objective function we maximise is the following

$$
\begin{aligned}
\arg\max_{\theta,\phi} \quad & \mathbb{E}_{q_\phi(x,\beta|y,\alpha)} \log p(y|x,\alpha,\beta) \\
& + \gamma \mathbb{E}_{q_\phi(x|y,\alpha)} \big[ \mathbb{E}_{q_{\phi_3}(z_p|x)} \log p_\theta(x|z_p) - KL(q_{\phi_3}(z_p|x)||p(z_p)) \big] \\
& + H(q_{\phi_1}(z|y,\alpha)) + \gamma \mathbb{E}_{q_{\phi_1}(z|y,\alpha)} H(q_{\phi_2}(x|z,y,\alpha)) \\
& - \lambda \left| \mathbb{E}_{q_\phi(z,x|y,\alpha)} \log \frac{r_{\phi_4}(z|x)}{q_{\phi_1}(z|y,\alpha)} - C \right|,
\end{aligned}
\tag{22}
$$

$q_\phi(x,\beta|y,\alpha) = \int_z q_{\phi_1}(z|y,\alpha)q_{\phi_2}(x|z,y,\alpha)q_{\phi_5}(\beta|z,y,\alpha)dz$, $\quad q_\phi(x|y,\alpha) = \int_z q_{\phi_1}(z|y,\alpha)q_{\phi_2}(x|z,y,\alpha)dz$, $q_\phi(z,x|y,\alpha) = q_{\phi_1}(z|y,\alpha)q_{\phi_2}(x|z,y,\alpha)$, $\phi = \{\phi_{1:5}\}$ are the parameters of the inference models and $\theta$ are the parameter of the prior model.

### B.3 PSEUDO-CODE

### B.4 POSTERIOR COLLAPSE IN VARIATIONAL INFERENCE

The posterior collapse problem we approach with the reduced entropy condition method presented in this paper has some analogy with the latent posterior collapse encountered when using implicit distributions in variational inference to obtain flexible recognition models. The main issue in training these models successfully without collapse is the computation of density rations between the latent prior and the implicit variational posterior. This problem is analogous to the difficulty in estimating the LVM entropy in our method. Yin & Zhou (2018) proposed to use a further lower bound on the ELBO and add a term encouraging diversity to avoid collapse. This term is obtained by drawing $K$ Gaussian components from the LVM posterior and computing the KL divergence of an individual component with the mixture distribution, i.e. the sum of the drawn Gaussians. Titsias & Ruiz (2019) build on this work by deriving an unbiased estimator for the ELBO gradient, instead of using a surrogate lower bound.

These methods rely on estimating the LVM posterior through sampling and aggregating $K$ explicit distribution components. This was demonstrated to work well for posteriors in artificial latent space by drawing only a few components. However, in our data recovery setting, we need to capture the posteriors in clean data space, rather than a latent space. Posteriors capturing the uncertainty in natural data are expected to be much more complex and higher dimensional, leading to the number $K$ of drawn Gaussians needed to approximate the true LVM with these methods to become rather large, making optimisation inefficient or even intractable in extreme cases. The reduced entropy condition method we derive in this paper avoids the posterior collapse without having to estimate the LVM posterior through sampling and is therefore specially suited for the data recovery setting, where we are required to capture posteriors in clean data space.

## C EXPERIMENTAL DETAILS

### C.1 MODELS' ARCHITECTURES

In all experiments we carry out comparing our TAE with competitive methods, we make the independence assumption $q(x|z,y) = q(x|z)$, consequentially making $r(z|x,y) = r(z|x)$. In this way, the reconstruction posterior LVMs $q(x|y)$ we compare between TAE, MVAE and MIWAE all present identical structure and differences in performance are a result of the model constructed to train them alone. However, we note that, unlike the two competing method, the TAE is not formally limited to this choice and can infer conditionals $q(x|z,y)$ in the general case. We hereafter detail the architecture used for all quantitative experiments of section 4.1 and 4.2.

**MVAE and MIWAE models:** The MVAE model is built as an LVM having a unit Gaussian prior in the latent space $p(z) = \mathcal{N}(z; \mathbf{0}, \mathbf{1})$ and isotropic Gaussian clean data likelihood $p(x|z) = \mathcal{N}(x; \mu_x, \sigma_x^2)$, where the moments $\mu_x$ and $\sigma_x^2$ are outputs of a neural network having as input the latent variables $z$. Because we only observe corrupted data $y$, rather than clean data $x$, the recognition model $q(z|y)$ is conditioned on observed corrupted data $y$ and also has the form of an isotropic Gaussian $q(z|y) = \mathcal{N}(z; \mu_z, \sigma_z^2)$, where the moments $\mu_z$ and $\sigma_z^2$ are outputs of a

---

**Algorithm 1** Training the TAE Model

---

***Inputs:*** Corrupted observations $Y = \{y_{1:N}\}$; Observed Parameters $A = \{\alpha_{1:N}\}$ initial model parameters, $\{\theta^{(0)}, \phi^{(0)}\}$; user-defined posterior latent dimensionality, $J$; user-defined prior latent dimensionality, $J_p$; user-defined condition strength $\lambda$; user-defined condition parameter $C$; user-defined latent prior $p(z_p)$; user-defined initial warm-up coefficient $\gamma_0$; user-defined final warm-up coefficient $\gamma_f$; warm-up start $N_{w0}$; warm-up end $N_{wf}$; user-defined number of iterations, $N_{iter}$.

1: $\gamma^{(k=0)} \leftarrow \gamma_0$
2: **for** *the $k$'th iteration* **in** $[0 : N_{iter} - 1]$
3:    **for** *the $i$'th observation*
4:       $z_i \sim q_{\phi_1^{(k)}}(z|y_i, \alpha_i)$
5:       $x_i \sim q_{\phi_2^{(k)}}(x|z_i, y_i, \alpha_i)$
6:       $\beta_i \sim q_{\phi_5^{(k)}}(\beta|z_i, y_i, \alpha_i)$
7:       $z_{p,i} \sim q_{\phi_3^{(k)}}(z_p|x_i)$
8:       $\mathbf{E}_i^{(k)} \leftarrow \log p(y_i|x_i, \beta_i)$
9:       $\mathbf{P}_i^{(k)} \leftarrow \log p_{\theta^{(k)}}(x_i|z_{p,i})$
10:      $\mathbf{K}_i^{(k)} \leftarrow D_{KL}(q_{\phi_3^{(k)}}(z_p|x_i)||p(z_p))$
11:      $\mathbf{Hz}_i^{(k)} \leftarrow H(q_{\phi_1^{(k)}}(z|y_i, \alpha_i))$
12:      $\mathbf{Hx}_i^{(k)} \leftarrow H(q_{\phi_2^{(k)}}(x|z_i, y_i, \alpha_i))$
13:      $\mathbf{R}_i^{(k)} \leftarrow \log r_{\phi_4^{(k)}}(z_i|x_i, y_i, \alpha_i)$
14:      $\mathbf{Q}_i^{(k)} \leftarrow \log q_{\phi_1^{(k)}}(z_i|y_i, \alpha_i)$
15:    **end**

16:    $\mathbf{F}^{(k)} = \sum_i \left( \mathbf{E}_i^{(k)} + \gamma^{(k)} \left[ \mathbf{P}_i^{(k)} - \mathbf{K}_i^{(k)} + \mathbf{Hx}_i^{(k)} \right] \right.$

17:       $\left. + \mathbf{Hz}_i^{(k)} - \lambda \left| \mathbf{R}_i^{(k)} - \mathbf{Q}_i^{(k)} - C \right| \right)$

18:    $\theta^{(k+1)}, \phi^{(k+1)} \leftarrow \arg\max(\mathbf{F}^{(k)})$

19:    **if** $k > N_{w0}$ **and** $k < N_{wf}$
20:      $\gamma^{(k+1)} \leftarrow \gamma^{(k)} + (\gamma_f - \gamma_0)/(N_{wf} - N_{w0})$
21:    **else**
22:      $\gamma^{(k+1)} \leftarrow \gamma^{(k)}$
23:    **end**
24: **end**

---

neural network having as input the corrupted observations $y$. The corrupt data likelihood $p(y|z)$ is obtained by simply selecting the likelihood $p(x|z)$ over the observed indexes, i.e. for missing values corruption the integral $p(y|z) = \int p(x|z)p(y|x)dx$ simply masks out the unobserved entries. The model is then trained by maximising the ELBO of equation 1. The MIWAE is built with the same structure, but instead of optimising the MVAE ELBO of equation 1, an importance weighted lower bound is maximised, as described in (Mattei & Frellsen, 2019). The precise architectures used for the neural networks are described for the different experiments throughout the rest of this section. One important point to notice is that, in each experiments, the structures of $p(x|z)$ and $q(z|y)$ are chosen such that the resulting reconstruction model after training, i.e. the model taking as input a test corrupt observation $y$ and generating clean samples $x$, is identical for the TAE and the two tested competing models. That is $\int q(z|y)p(x|z)dz$ for the MVAE and MIWAE have identical structure to $\int q(z|y)q(x|z)dz$ for the TAE. In this way, performance differences can be attributed solely to the difference in inference strategy and not reconstruction model's capacity.

**Posteriors structure:** The posterior parametric components are $q_{\phi_1}(z|y,\alpha)$ and $q_{\phi_2}(x|z)$ $(p_{\phi_2}(x|z)$ in the case of the MVAE and MIWAE). $q_{\phi_1}(z|y,\alpha)$ consists in a fully connected two layers neural network with leaky ReLu non-linearities, taking as input concatenated corrupted observations $y$ and a binary mask that labels the missing entries $\alpha$ and returning as output a vector of latent means and a vector of latent log variances. The two intermediate deterministic layers have $400$ hidden units, while the latent space $z$ is 20-dimensional.

$q_{\phi_2}(x|z)$, and $p_{\phi_2}(x|z)$ in the case of the MVAE and MIWAE, are similarly constructed, consisting in a fully connected two layers neural network with leaky ReLu non-linearities, taking as input latent variables $z$ and returning a vector of means and a vector of log variances of clean samples $x$. The two intermediate deterministic layers have $400$ hidden units.

**TAE Prior LVM Structure:** The TAE prior encoder $q_{\phi_3}(z_p|x)$ has the same general structure as the posterior encoder, with two fully connected layers and leaky ReLu non-linearities, taking as input generated clean data $x$ and returning as outputs a vector of latent means and a vector of latent log variances for the prior latent variable $z_p$. As this model has less capacity than the posterior LVM, the two deterministic hidden layers have $50$ hidden units each and the latent variables $z_p$ are 5-dimensional.

$p_\theta(x|z_p)$ is similarly constructed, consisting in a fully connected two layers neural network with leaky ReLu non-linearities, taking as input latent variables $z_p$ and returning a vector of means and a vector of log variances of clean samples $x$. The two intermediate deterministic layers have $50$ hidden units.

**Approximate Latent Posterior Structure:** The approximate latent posterior $r(z|x)$ has the same structure as the posterior encoder, consisting in a fully connected two layers neural network with leaky ReLu non-linearities, taking as input generated clean data $x$ and returning as outputs a vector of latent means and a vector of latent log variances. The two intermediate deterministic layers have $400$ hidden units.

**Convolutional TAE Structure:** For the imputation of NYU missing data we use convolutional conditionals in our model, instead of fully connected ones. In this version. we do not make the independence assumption, using $q(x|z,y)$ and $r(z|x,y)$. $q(z|y,\alpha)$ takes concatenated $y$ and $\alpha$ and passes them through $4$ recurrent convolutional layers with filters of size $3 \times 3$ and $5$ channels, each time down-sampling by two. the last layer is mapped to means and standard deviation of latent images $z$, which are $1/32$ of the original size in each axis and have $10$ channels, through two convolutional filter banks with strieds $1 \times 1$. $q(x|z,y,\alpha)$ is built to mirror this structure, with the addition of accepting inputs from $y$ and $\alpha$. Three recurrent transpose convolutional layers with $3 \times 3$ filters, $5$ channels and $2 \times 2$ upsampling each map $z$ to a deterministic layer with $1/2 \times 1/2$ of the original images size. concatenated $y$ and $\alpha$ are mapped to the same size with a single convolutional layer, downsampling it by $1/2 \times 1/2$ and $5$ channels. The two are concatenated and the resulting layer is finally upsampled to inferred clean image $x$ by a last convolution with a filter bank. All non-linearities are Elu.

The prior networks are built in a similar way, but with shallower structures to give less capacity. $q(z_p|x)$ passes $x$ through $2$ convolutional layers, each with down-sampling of $4 \times 4$ and $5$ channels. as before, mens and standard deviations of latent images $z_p$ are generated from this last layers with $2 \times 2$ down-sampling and, in this case, $5$ channels. The prior generator $p(x|z_p)$ is built to exactly mirror this structure. $r(z|x,y,\alpha)$ has the same structure as $q(z|y,\alpha)$, with the only difference being that it accepts as input concatenated $x$, $y$ and $\alpha$.

## C.2 EXPERIMENTAL CONDITIONS

**Posterior Recovery:** All posterior recovery experiments, with each of the three data sets tested, are performed on samples that have been re-scaled from $0$ to $1$. In all cases, the sets are injected with additive Gaussian noise having standard deviation $0.1$. Subsequently, random binary masks are generated to block out some entries, resulting in missing values. The proportion of missing entries in the masks was set as described in the main body in each case.

Experiments were repeated with re-generated binary masks 5 times. The means and error bars shown in figure 4 and the uncertainty reported in table 1 were computed from these. The MIWAE was trained with 20 weights per sample. After training, all posteriors $q(x|y)$ have identical structure and

are tested in the same way, by training an inference network on the test set to compute the ELBO values.

**Classification Experiments:** The TAE models for the MNIST and Fashion-MNIST experiments were trained in the conditions described above. In each case, a random subset of $10,000$ samples is taken from the corrupted set and the TAE and MVAE models are trained with it. A random subset of $1,000$ of these is selected and ground-truth lables for these samples are made available.

A classifier consisting in a single fully connected layer with leaky ReLu non-linearity is trained to perform classification on this subset. For each stochastic training iteration of this classifier, we generate samples associated with the corrupted observations and provide the associated labels. After the classifier is trained, we test classification performance on the remaining $9,000$ examples, by running the train classifier $400$ times per sample, each time generating clean data from a corrupted observation with the TAE and the MVAE. The histograms shown in figure 5 are built by aggregating the resulting classification.

The above procedure is repeated $15$ times. The resulting means and standard deviations of the tested classification performance are shown in figure 6.

**Training Conditions:** Hyper-parameters of optimisation for the models were cross validated with the MNIST data set at a proportion of missing entries of $0.9$. Hyper-parameters common to all models were determined by obtaining best performance with the MVAE model. Hyper- parameters specific to the TAE model were obtained by fixing the common parameters and cross validating these. The resulting optimal hyper parameters were then used in all other experiments of section 4.1 and 4.2, including those with different data sets. Common parameters are as follows: $500,000$ iterations with the ADAM optimiser in Tensorflow, an initial training of $2^{-4}$ and batch size of $20$. The hyper-parameters specific to the TAE are instead: $\gamma$ initially set to $0.01$ and then linearly increased to $1$ between $50,000$ and $100,000$ iterations, $\lambda = 2$ and $C = 10$. All experiments were performed using a TitanX GPU.

**NYU Rooms Experiments:** For these experiments, we take a subset of $3612$ depth maps from the NYU raw data set. We slightly crop these in one dimension to be $480 \times 608$ images. The convolutional TAE and MVAE to obtain the results of figure 7, were trained for $100,000$ iterations using the ADAM optimiser in Tensorflow, with a batch size of $20$ images and an initial training rate of $2 \times 10^{-2}$. For the warm up, we initially set $\gamma = 0.01$ and linearly increase it to $1$ between $10,000$ and $20,000$ iterations. For these experiments, $\lambda = 2$ and $C = 15$.

### C.3 Evaluation ELBO

To evaluate the probabilistic performance of our method compared to others, we compute an evaluation ELBO which relies on test ground truths. After each model is trained unsupervisedly, we obtain a posterior of the form $q(x|y) = \int q(z|y)q(x|z)dz$, where for the MVAE and MIWAE, $q(x|z) = p(x|z)$. Given the a test set of paired clean and corrupted samples $x_t$ and $y_t$, we construct a new parametric recognition model, which encodes latent distributions from ground-truths $q_\eta(z|x)$. We then optimise the following:

$$\arg\max_\eta \quad \mathbb{E}_{q_\eta(z|x_t)} \log q(x_t|z) + KL(q_\eta(z|x_t)|q(z|y_t)). \tag{23}$$

The above is a conditional VAE ELBO with conditional prior $q(z|y)$ and is a lower bound to the test likelihood we are interested in $q(x_t|y_t)$. Note that we optimise over $\eta$ only, therefore the new recognition model $q(z|x)$ is the only one which is affected by this optimisation and the components of our reconstruction model $q(z|y)$ and $q(x|z)$ remain the same as trained with the unsupervised training set. As a result, this new optimisation only tightens the bound, rather than maximising the likelihood, which we want to evaluate as trained previously.

## D Additional Experiments

### D.1 $C$ and $\lambda$ Cross-Validation

$C$ and $\lambda$ in equation 22 are hyper-parameters of our inference algorithm and need to be user defined. In our experiments, we determine the optimal values by cross-validation, as described in section C.

We report in figure 7 a cross validation study where we measure the TAE ELBO for MNIST with 90% missing values and additive noise.

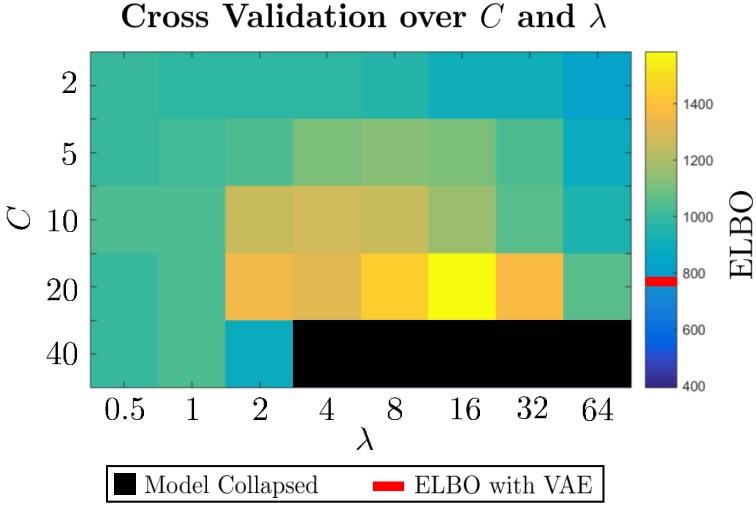

Figure 7: ELBO for MNIST with 90% missing values and additive noise as a function of chosen hyper-parameters $C$ and $\lambda$ (in log scale). The performance of TAE exceeds that of a standard VAE approach over a broad range of values. If the values are too large, the model collapses during optimisation, making such situation easy to diagnose.

As shown in figure 7, the performance of TAEs is robust to variations in hyper-parameters $C$ and $\lambda$ over a broad range of values. They also have an intuitive meaning that helps in their selection. In practice, $C$ controls the final value of localisation and is desirable to be as high as stability of the optimisation allows. $\lambda$ controls how fast we are imposing the model to approach $C$.

### D.2 STRUCTURED MISSINGS

We test a TAE in a situation analogous to that shown in figure 4 of section 4, but with structured missing values instead of randomly missing ones. For each sample in MNIST, we only make visible a small $10 \times 10$ pixels window, randomly placed in each example, while the rest of the images remain hidden. In addition, the values in the observed window are subject to additive Gaussian noise, similarly to the missing-at-random case. Reconstructions with the comparative MVAE and our TAE are shown in figure 8.

Similarly to the missing-at-random case, the MVAE collapses on single solutions, giving draws from the posterior that are all very similar to each other. Contrarily, the TAE gives more variation in the possible solutions exploring more appropriately the uncertainty in the solution space. The MVAE ELBO over the clean data for this problem is 428, while the TAE one is 638. The performance improvement provided by the TAE is analogous to that observed with missing-at-random experiments.

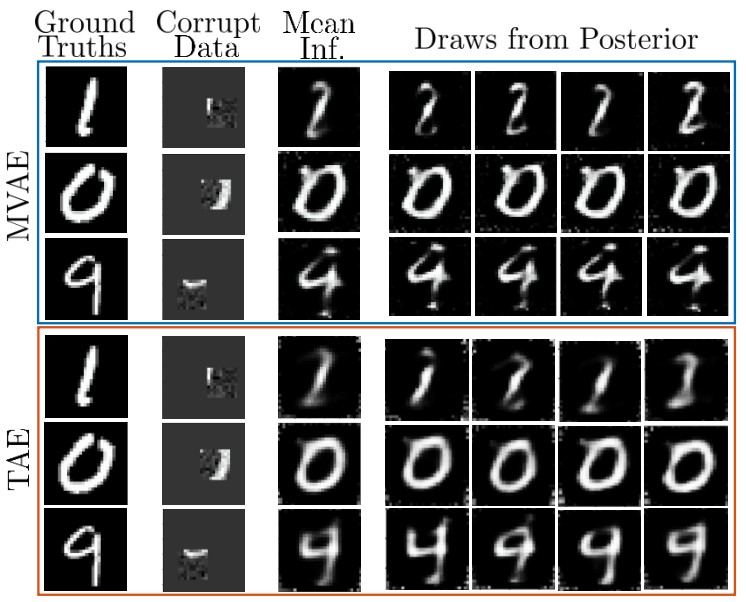

Figure 8: Examples of Bayesian reconstructions with MVAE and TAE on structured missing values. the MVAE returns good mean reconstructions, but its posteriors collapse on single solutions, giving draws that are very similar to each other. The TAE returns posteriors which more broadly explore the different possible clean samples associated with the corrupted observations, giving more variation in the posterior's draws.

## D.3 MORE ELBO EVALUATIONS

Table 2: ELBO assigned by the retrieved posteriors to the ground truth clean data.

|  | MVAE | MIWAE | TAE |
|---|---|---|---|
| MNIST, 20% missing | $883 \pm 2$ | $940 \pm 3$ | $\mathbf{1831 \pm 8}$ |
| MNIST, 50% missing | $870 \pm 6$ | $917 \pm 4$ | $\mathbf{1719 \pm 7}$ |
| MNIST, 80% missing | $803 \pm 15$ | $780 \pm 6$ | $\mathbf{1536 \pm 14}$ |
| Fashion-MNIST, 20% missing | $775 \pm 4$ | $815 \pm 4$ | $\mathbf{1407 \pm 24}$ |
| Fashion-MNIST, 50% missing | $757 \pm 1$ | $800 \pm 7$ | $\mathbf{1326 \pm 7}$ |
| Fashion-MNIST, 80% missing | $723 \pm 7$ | $766 \pm 8$ | $\mathbf{1094 \pm 13}$ |
| UCI HAR, 20% missing | $611 \pm 3$ | $628 \pm 10$ | $\mathbf{1039 \pm 11}$ |
| UCI HAR, 50% missing | $585 \pm 4$ | $613 \pm 6$ | $\mathbf{1014 \pm 6}$ |
| UCI HAR, 80% missing | $471 \pm 10$ | $584 \pm 8$ | $\mathbf{854 \pm 52}$ |

## D.4 IMPUTATION WITHOUT NOISE

We carry out experiments on MNIST analogous to those shown in figure 3, but in the absence of noise, in order to test performance on imputation alone. Each tested ratio of observed entries is

repeated three times, re-generating the patterns of missings each repeat in order to obtain error bars. Results are shown in figure 10.

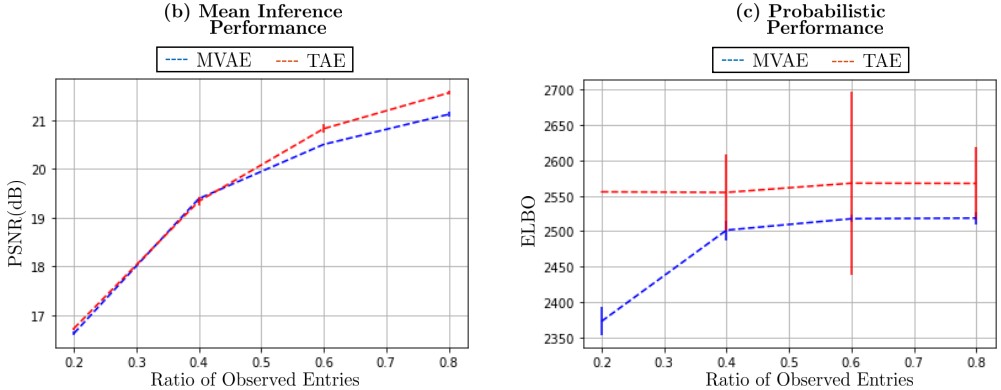

Figure 9: Missing value imputation performance on MNIST in the absence of noise. As in the noisy case, the PSNR values between the MVAE and the TAE are very similar. The TAE presents significantly superior ELBO values at low ratios of observed entries, but in this case, the gap is reduced as more entries are observed. This is because in the noiseless case, the solution space when most entries are observed is much more localised than in the noisy case, and therefore the MVAE collapsed posteriors do not fail as much to capture it.

## D.5 DE-NOISING

We carry out experiments on MNIST analogous to those shown in figure 3, but testing fully observed images at different levels of noise. Each tested ratio of observed entries is repeated three times, re-generating the patterns of missings each repeat in order to obtain error bars. Results are shown in figure 10.

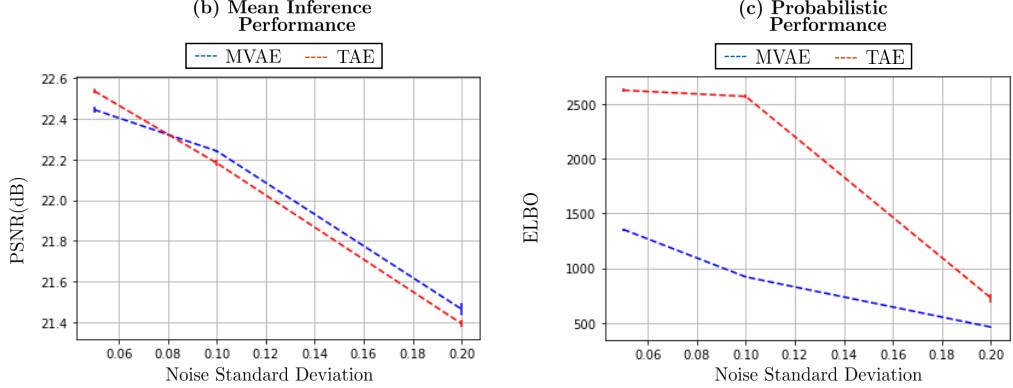

Figure 10: De-noising performance on MNIST. As in the missing value imputtion case, the $MVAE$ and $TAE$ perform very similarly in their mean reconstructions, but the TAE presents significantly better performance in capturing the distributions of clean solutions, as the test ELBO values are higher.

## D.6 NYU ROOMS RECOVERY EXAMPLES

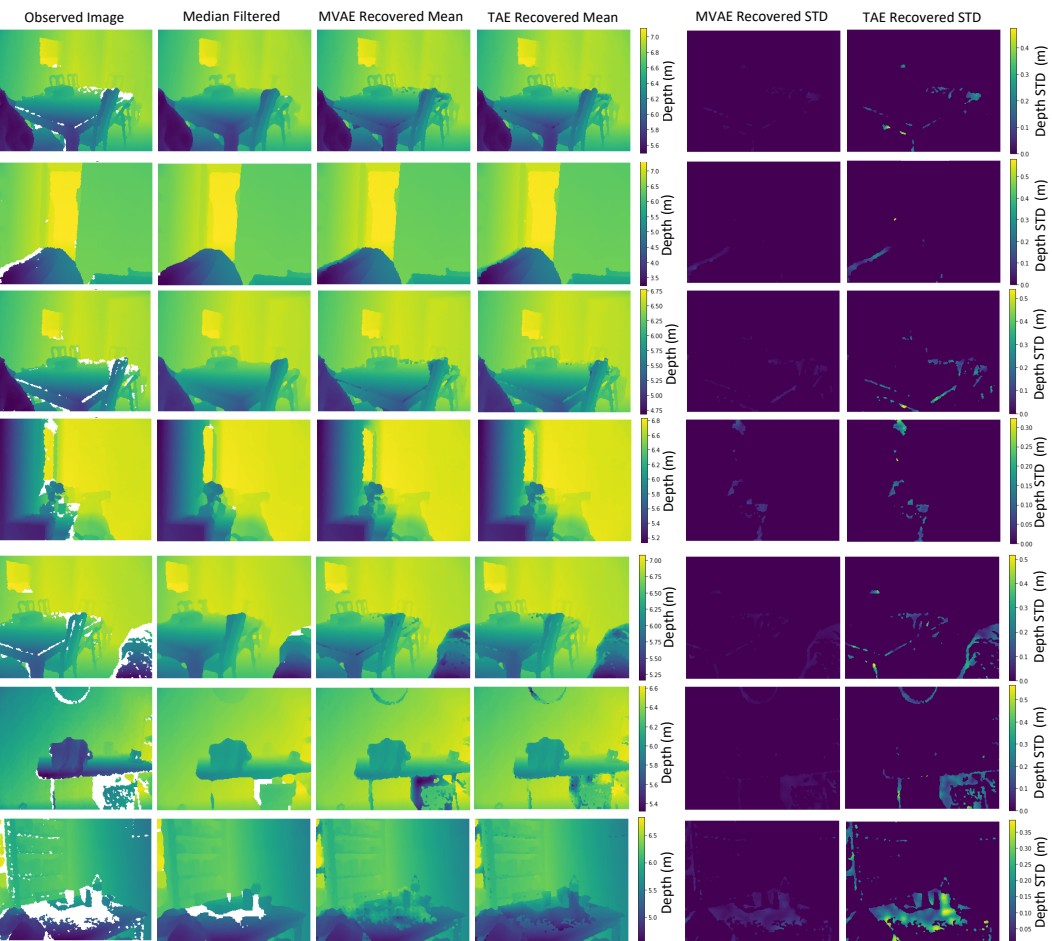

Figure 11: Unsupervised missing value imputation with our TAE on raw depth maps from the NYU rooms data set, compared with a median filter approach and the standard MVAE. Missing pixels in the observed images are in white. The median filter results in overly smoothed images and is unable to fill pixels that are surrounded by large missing areas. The MVAE returns adequate reconstructions, however, it over-fits in certain locations and its uncertainty is largely over-estimated. The TAE returns good reconstructions and assigns high uncertainty to locations where reconstruction is most inaccurate, as shown by the marginal standard deviations.

