# OpenReview forum: "Tomographic Auto-Encoder: Unsupervised Bayesian Recovery of Corrupted Data"
_ICLR.cc/2021/Conference — ICLR 2021 Poster_

### Official Review · AnonReviewer4 · 2020-10-28
**An interesting model for unsupervised recovery of corrupted data**

**Rating:** 7
**Confidence:** 3

**Review:**

Practical datasets often come with corruptions, such as missing items or noisy observations, thus needs models enable to recover the corrupted data automatically. This paper presents the tomographic auto-encoder (TVAE), which conducts inference over the data space $x$. Because the prior regularization acts over the data space, TVAE is enforced to generate diverse samples from the corrupted observations. Empirically, the paper demonstrates that TVAE can indeed generate diverse samples and can achieve superior test ELBO compared to the previous baselines.

Missing data imputation is one important problem in machine learning, given the imperfection of the practical data. Specifically, TVAE focuses on two properties: 1) unsupervised learning, which resolves the intractability of labelling large datasets 2) diverse recoveries, which attempts to generate multiple possible samples instead of collapsing onto one possibility. Towards these two properties, TVAE proposes to conduct inference over the data space $x$, whose prior directly prevents from collapsing.  To resolve the intractability of the entropy $H(q(x|y))$, TVAE identifies the *reduced entropy condition* and transforms the intractable ELBO maximization into a constrained optimization problem. The resulting model demonstrates superior performances compared to baselines, in terms of generating diverse samples. I think this work makes an important contribution.

This paper is well written.

Conditional neural process (Garnelo et. al., 2018a) and neural process (Garnelo et. al., 2018b) are another stream of models for missing data imputation, which are missed by the paper.

Regarding the reduced entropy condition, though it ensures that the entropy is decomposed into more tractable forms, will enforcing this condition limit the expressiveness of the encoder $q(x|  y)$ ?

---

> ### Author Response · Authors · 2020-11-17
> **Response to Review**
>
> We thank the reviewer for their review and for pointing out the relevant references references. We have now added to the related work section 3.2.
>
> As for the question about the expressiveness of TAE given the reduced entropy condition; in principle, expressiveness is not limited by the condition. Essentially, the reduced entropy condition imposes the conditional LVM to present a one-to-many mapping from latents z to clean data x. This simply prevents multiple z from contributing to the density in the same clean data point x, i.e. it promotes non-overlapping likelihoods. Since we avoid such degeneracies, we argue that the expressiveness of the model is improved, as we exploit all of the capacity to build expressive distributions.
> In practice, we are only approximately imposing the condition through the constrained minimisation, so the effects on expressiveness are difficult to define analytically. What we can say is that experimentally training q(x|y) with the TAE method resulted in reconstruction models which were not any less expressive than identical ones trained with the MVAE in their mean inferences, as they gave similar average PSNR over a test set. Instead, the individual posteriors q(x|y_i) were found to be significantly more expressive with the TAE, as shown by the ELBO values and samples in figure 3.

---

### Official Review · AnonReviewer2 · 2020-10-28
**Interesting VAE variant for reconstructing noise-free versions of corrupted observations without supervision**

**Rating:** 7
**Confidence:** 3

**Review:**

Summary: The paper proposes a method for reconstructing noise-free data instances without assuming any ground truths for this, using a variant of auto-encoder that avoids posterior collapse by utilising a newly proposed reduced entropy condition. The problem itself is important and the proposed method seems to offer good empirical performance for the task, outperforming a good selection of recent methods. The paper is well written with ample citations for relevant work, and provides sufficient technical details.

Reasons for score: A good paper with no obvious flaws; I do not have concrete improvement suggestions.

Additional comments: The key concept of empirical prior and its use for regularising the autoencoder is good and intuitive, and I could not spot any obvious theoretical or practical issues with the idea. In the end the theoretical development results in fairly simple modification for the standard VAE objective that can be directly trained using standard algorithms. This is both a strength (the approach is easy to implement and can be plugged into existing models) and a minor weakness in terms of technical depth. Nevertheless, the justification for the loss is well explained in the paper.

The empirical experiments are sufficiently comprehensive and show good empirical performance in a range of tasks, against reasonable comparison methods. As a minor comment, I would recommend using a more consistent style for the figures (e.g. Fig 3 and Fig 5 use very different visual style).

---

> ### Author Response · Authors · 2020-11-17
> **Response to Review**
>
> We thank the reviewer for their comments and we are happy to see that the paper was very well receive. we respond to specific points below.
>
> The final objective function we present is indeed relatively simple to implement, which, as pointed out by the reviewer, is an advantage as the TAE can be trained with standard approaches. However, we point out that this objective is not motivated by heuristics, and its derivation with the associated formulation of the reduced entropy condition are far from obvious. The full derivation is presented over several pages in the supplementary A2-A5. We argue that this new approach provides considerable technical depth while resulting in an easily implementable algorithm.
>
> We thank the reviewer for the suggestion. We tried to arrange results in the experiments’ figures to both be clear and as compact as possible for space limitations. As additional space will be given for the final version of the paper, we will revisit the images such that the style across the experimental results is more consistent.

---

### Official Review · AnonReviewer3 · 2020-10-28
**Good paper on corrupted data, although mainly focused on missing data**

**Rating:** 7
**Confidence:** 4

**Review:**

Review:

This paper proposes a novel approach to handle the recovery of dirty data in fully unsupervised scenarios. The corrupted data considers both missing data and noisy samples. They derive a VAE model with a novel reduced entropy condition inference method that results in richer posteriors. This is a very challenging problem, since the model cannot use clean examples as part of their training procedure.

Questions/Comments:

- It feels to me that the main focus of this paper is on missing data rather than in other types of corruption. I get this impression mainly from the experimental section and methods used for comparison, well known for handling missing data. Something I would have appreciated in this work is to observe the performance of the authors method in scenarios handling missing data and corrupted data separately. Nonetheless, it is interesting to see experiments with both effects combined, which is not so common in the literature.

- Would it be possible to get results beyond maximizing the ELBO? The ELBO contains the reconstruction term of the images, but also many other terms. In the end, in a missing data imputation model, it is interesting to get an idea of how good the reconstruction of the images is. I can see from figures 3 and 4 that they should be good, but having a different metric might be helpful. I find it surprising that in Table 1 the ELBO of TAE is almost double of the other methods, while the average reconstruction of MVAE for example does not look that much different in figure 3 from the ground truth, and it is pretty similar to TAE.

Summary:

The paper is well written and the idea is novel as far as I know. The notation is clear and the proofs in the appendix look sensible. The experimental section showcases several scenarios where they compare to unsupervised generative models to handle missing data. The analysis of corrupted data (outside of missing data) seems a bit lacking.

---

> ### Author Response · Authors · 2020-11-17
> **Response to Review**
>
> We thank the reviewer for their feedback and answer below to the main comments:
>
> - We welcome the suggestion of testing separately de-noising and missing value imputation. For the final version of the paper, we will include experiments analogous to those shown in figure 3 repeated with no noise injection and will extend the graphs to a ratio of observed entries of 1, hence covering both situations.
>
> - Figure 3b shows the peak signal to noise ratio (PSNR) between the mean reconstruction and the ground truth image, which is a measure of average reconstruction quality. The PSNR is very similar for the MVAE and TAE. As observed by the reviewer, the mean reconstructions look indeed of similar quality, and the PSNR values reflect this.
> The ELBO we use is calculated by training a new inference model with the clean data (described in supplementary C.3) and is therefore approximating reconstruction log likelihood. The reason for the mean reconstruction quality and the log likelihood being so different is the collapse; the MVAE finds a fitting solution to the reconstruction problem, but greatly underestimates uncertainty, so the mean performance is good, while the probabilistic performance, measured by the ELBO, is low. The TAE, instead, captures well the full distribution of possible reconstructions, giving much higher ELBOs, although the mean reconstruction is of similar quality.

---

### Official Review · AnonReviewer1 · 2020-10-30
**Good empirical performance but questionable model design**

**Rating:** 7
**Confidence:** 4

**Review:**

This paper proposes a Tomographic auto-encoder (TAE) for unsupervised recovery of corrupted data. More specifically, TAE takes a Bayesian approach to recover the posterior distribution of a clean image conditioned on an observed corrupted image and thus effectively modeling uncertainty in data recovery. The paper argues that a naive application of VAE is not effective due to the latent variable collapse, and proposes an alternative model where hierarchical latent variable models are used for both prior and variational posterior. Some tricks are introduced to facilitate the stochastic gradient variational inference.

I think the paper is tackling an important problem and I advocate the use of a VAE-like model for uncertainty modeling. The paper is clearly written with helpful figures. The experimental results, at least compared to the baseline (MVAE), looks promising. I like the way the methods are compared using the downstream task.

However, I'm not sure whether a new model should be developed besides the existing approaches. The authors state that the problem of a vanilla VAE with hierarchical latent structure (latent code $z$ - clean image $x$) is that it is prone to latent variable collapse.  This is true, but there are plenty of existing works (partially) resolving this problem. The most relevant approach I can think of is the semi-implicit variational inference [1] for which a hierarchical latent variable model is used for the variational distribution similar to the setting considered in this paper. [1] proposes a theoretically guaranteed solution to prevent a trivial case where the lower-level latent variable ($z$) completely collapses into a point mass, and I think this can directly be applied for the problem considered in this paper. Also, as [2] pointed out that one can consider using a more expressive prior distribution for latent code $z$ to combat latent variable collapse. For instance, flow-based models may be employed for both prior and variational posterior. The proposed design also makes sense, but I don't think they are more expressive than the models listed above. If that is the case, there should be a special factor making TAE specifically well-suited for the problem at hand - the recovery of corrupted images - but I failed to find such a thing. Is there any reason that the existing models other than the vanilla VAE cannot be considered?

Also, the baseline (MVAE) is not clearly described, so it is not clear how MVAE was actually implemented. I recommend giving a more detailed description of the baseline at least in the appendix.

References
[1] Yin and Zhou, Semi-implicit variational inference, ICML 2018.
[2] Chen et al., Variational lossy autoencoder, ICLR 2016.

---

> ### Author Response · Authors · 2020-11-17
> **Response to Review**
>
> We thank the reviewer for their comment and are happy to see that they agree with the scope of our work. We respond below to the main concern regarding the posterior collapse.
>
> We do not state that the problem is collapse in the latent space (z). This is not any more of a problem in corrupt data recovery than it is in standard VAE settings. The problem is the collapse in clean data space (x).
>
> The posteriors in z given by the MVAE are not critically collapsed [1,2], it is the reconstruction posteriors in x that are severely collapsed. This is because the recognition model encodes from partial information (the corrupted data y), but then we aim to reconstruct the complete information (the clean data x), differently from vanilla VAEs which encode from and reconstruct the same inputs. MVAEs and other models, which do not regularise in x space, are free to reconstruct a single answer x that explains an observation y, instead of exploring the full distribution. This occurs even though latent space posteriors are not collapsed. A more complex prior in z does not solve this problem.
>
> There is an analogy between the posterior collapse encountered in latent spaces in variational inference (VI) and the collapse encountered in clean data space in our settings, but the two present critically different challenges. In practice, the difference is the dimensionality and complexity of the posteriors we need to model. Semi-implicit VI and other related works avoid the collapse by drawing several samples K from the posterior model and using them to encourage diversity in some way. For instance, semi-implicit VI computes a mixture of K Gaussians and maximises the KL of each individual component with the mixture. These methods rely on the discrete aggregate (the mixture) approximating well the true model, which for latent spaces, as demonstrated by these works, is achieved with a few samples (K~10), keeping the optimisation relatively efficient. This is not the case in the data recovery case we focus on, where the posteriors need to represent natural data distributions in high dimensions (several hundreds to several thousands of dimensions in our examples). The number of components K we would need to approximate well the true posterior scales with both the dimensionality and the complexity of the true posterior, which are both very high in our target scenarios.
>
> Given the above, the corrupted data setting presents a unique challenge: how can we avoid the collapse without relying on the prohibitive sampling of multiple components of our LVM posterior to approximate the true distribution? This is what the reduced entropy condition method achieves. So in answer to the reviewer’s question: “Is there any reason that the existing models other than the vanilla VAE cannot be considered?” - yes, the reason is that avoiding the posterior collapse in data space (x) rather than a latent space (z) requires being more efficient with respect to sampling components from the posterior LVM than existing collapse-avoiding methods.
>
> We briefly state this difference before explaining the reduced entropy condition on page 4 and cite a more recent work related to the original semi-implicit VI [3]. We do appreciate that this discussion may deserve more depthening and thank the reviewer for their comment and references. We have added a subsection covering the relevant literature and explaining this difference in the supplementary (B.4). We will consider moving this to the related work section in the main body, given the extra page.
>
> refs:
> [1] Nazabal et al. Handling incomplete heterogeneous data using VAEs. arXiv preprint. 2018.
> [2] Dalca et al. Unsupervised Data Imputation via Variational Inference of Deep Subspaces. arXiv preprint. 2019.
> [3] Michalis K. Titsias and Francisco Ruiz.   Unbiased implicit variational inference. AISTATs 2019.
>
> As for the MVAE details, we thank the reviewer for the recommendation and we have added a subsection in supplementary C.1 describing in more detail the structure of the competing MVAE.

---

> > ### Comment · AnonReviewer1 · 2020-11-19
> > **Thanks for clarification!**
> >
> > Dear authors,
> > Thanks for the clarification and sorry for my misunderstandings. Now my concern is resolved so I raise my score.

---

### Author Response · Authors · 2020-11-17
**Authors' Response**

We thank the reviewers for their thoughtful feedback. We are glad to see that our paper was generally well received and appreciated. We have replied with detailed responses to each reviewer below.

 In particular, we feel that the central point raised by reviewer 1 might be a result of a misunderstanding concerning the posterior collapse problem, which constitutes the central technical challenge related to our approach. In short, it is not the collapse of the latent variables z that we aim to solve, but the collapse in data space x. This is a problem unique to data recovery, i.e. not encountered in standard VAE models. We elaborate below in the response to reviewer 1.

---

### Decision · Program_Chairs · 2021-01-07
**Final Decision**

**Decision:**

Accept (Poster)

**Comment:**

Summary:
The authors propose a Bayesian approach to data cleaning, implemented
via a variational auto-encoder. They argue that a common problem in
this context are posteriors that overfit by
concentrating on a low-dimensional subset and introduce an
optimization target intended to discourage that behavior.

Discussion:
Arguably the main concern brought up in the reviews was how
much novelty there is in addressing latent variable posterior
collapse, solutions for which have been proposed. The authors were able to clarify that this was due to a
misunderstanding (the collapse they address is not in latent space),
and the reviewer considers the matter resolved.


Recommendation:
I recommend publication. The reviewers are all
positive, agree that the method is interesting, and seems novel. The
writing is clear, and remaining doubts have been addressed in the discussion.